# Improving the Sexual Health of Young People (under 25) in High-Risk Populations: A Systematic Review of Behavioural and Psychosocial Interventions

**DOI:** 10.3390/ijerph18179063

**Published:** 2021-08-27

**Authors:** Ellie Brown, Samantha Lo Monaco, Brian O’Donoghue, Hayley Nolan, Elizabeth Hughes, Melissa Graham, Magenta Simmons, Richard Gray

**Affiliations:** 1Orygen, 35 Poplar Road, Parkville, VIC 3052, Australia; samantha.lomonaco@unimelb.edu.au (S.L.M.); Brian.odonoghue@unimelb.edu.au (B.O.); Hayley.nolan@unimelb.edu.au (H.N.); msimmons@unimelb.edu.au (M.S.); 2Centre for Youth Mental Health, The University of Melbourne, 35 Poplar Road, Parkville, VIC 3052, Australia; 3School of Healthcare University of Leeds, Woodhouse, Leeds LS2 9JT, UK; E.C.Hughes@leeds.ac.uk; 4College of Science, Health and Engineering, La Trobe University, Bundoora, VIC 3086, Australia; m.graham3@latrobe.edu.au (M.G.); R.gray@latrobe.edu.au (R.G.)

**Keywords:** adolescents, sexual health, high risk youth, at risk populations, young people

## Abstract

*Background:* Ensuring young people experience good sexual health is a key public health concern, yet some vulnerable groups of young people remain at higher risk of poor sexual health. These individuals require additional support to achieve good sexual health but the best way to provide this remains needs to be better understood. *Methods:* We searched for randomised controlled trials of behavioural and psychosocial interventions aimed at promoting sexual health in high-risk young populations. Outcomes of interest were indicators of sexual health (e.g., condom use, attitudes to contraception, knowledge of risk). Participants were under 25 years old and in one of the following high-risk groups: alcohol and other drug use; ethnic minority; homeless; justice-involved; LGBTQI+; mental ill-health; or out-of-home care. *Results:* Twenty-eight papers from 26 trials met our inclusion criteria, with all but one conducted in North America. Condom use was the most frequently reported outcome measure along with knowledge and attitudes towards sexual health but considerable differences in measures used made comparisons across studies difficult. Change in knowledge and attitudes did not consistently result in long-term change in behaviours. *Conclusions:* There remains a dearth of research undertaken outside of North America across all high-risk groups of young people. Future interventions should address sexual health more broadly than just the absence of negative biological outcomes, with LGBTQI+, homeless and mental ill-health populations targeted for such work. An international consensus on outcome measures would support the research field going forward, making future meta-analyses possible.

## 1. Introduction

Good sexual health is broader than just being free from sexually transmitted infections (STIs). Rather, it is defined as experiencing sexuality that is satisfying, positive, and respectful, as well as being free from exploitation and violence [1]. This definition highlights how an individual’s sexual health is a function of complex and interconnecting biological, psychological, and social factors in their life. The consequences of poor sexual health are also multifaceted. Physical illness caused by bloodborne viruses (HIV, Hepatitis B and C) continue to contribute substantially to all-cause mortality [2,3], and infections transmitted during sex, such as human papillomavirus (HPV) is recognised as the main cause of cervical cancer [4]. While the psychological impact of, for example sexual assault, has also been widely discussed [5], what remains largely missing from the discourse is the general overlap between mental and sexual health [6,7]. By beginning to examine sexual health as an integral component to an individual’s wellbeing, negative physical and mental health consequences could potentially be mitigated. Furthermore, the positive outcomes associated with holistic sexual wellbeing could be enhanced. Like any area of public health, consideration should be given to good sexual health from an early age.

A critical period of time in an individual’s sexual development is the transition period between childhood and adulthood (typically considered between 10 and 25 years). This is a time of significant biological, social, and psychological changes from the start of puberty to sexual maturation. During this time sex hormones increase, bodily changes occur and a sense of self as a sexual being (including sexual identity) develops. Adolescents can experience newly emerging sexual desires and encounter opportunities to experiment with sexuality [8]; however, emotional maturity has often not yet fully developed. As a result, adolescents and young people are more likely to engage in what can be considered “high-risk” sexual activity. By this we mean activities that could lead to unintended pregnancy and/or sexually transmitted infections (STIs) such as condomless vaginal, anal, or oral sex, or sex while under the influence of drugs or alcohol [9]. This increased likelihood of engaging in high-risk sexual activity makes sexual health a critical health issue within this age group (under 25). This concern is supported by continued findings that a high proportion of new STIs and HIV infections occurred among 15–24-year-olds (for example, in the USA, these were as high as 50% for STIs [10], and 40% for new HIV infections [11]). Furthermore, when pregnancy occurs during adolescence rather than later in life, it is frequently associated with poor social outcomes [12].

In response to the need to address sexual health during this critical period, international agencies such as the United Nations Educational, Scientific and Cultural Organization (UNESCO) have declared that the delivery of sex education in schools should be mandatory [13]. Whilst these school-based programs ensure the delivery of sex education to many, it clearly continues to not do enough to improve the sexual wellbeing of all young people. There remain several groups of young people that experience additional vulnerabilities that place them at higher risk of poor sexual health, and it is likely that these individuals require a tailored approach that targets their specific needs. One such example is that of black American adolescents, with a recent meta-analysis finding that such sexual health interventions targeting this population successfully improved sexual health knowledge, attitudes and behaviour [14]. The authors concluded that there is a need for wide-scale dissemination of these programs to address racial disparities in sexual health across the US. This of course is just one ‘high-risk’ population of young people in one defined country and many other underserved populations exist.

When discussing ’high-risk’ groups it is important to acknowledge the impact this label can have on people, especially young people and those who are already marginalised by society for reasons such as ethnicity or socioeconomic status [15]. Therefore, in this review we have chosen to search for interventions targeting specific and defined populations of young people (under 25 years) who are underserved by standard sex education, focusing on definitions in the Australian Department of Health’s topic overview of ‘youth at risk’ [16]. This includes: individuals who use alcohol and other drugs (AOD), ethnic minorities other than black/African-American adolescents (given a recent review by Evans and colleagues [14]), young people experiencing homelessness, those in justice/prison or detention settings, LQBTQI+, including young men who have sex with men (YMSM), youth who experience mental ill-health, and those who are living in foster or out-of-home care (OOHC).

These groups represent populations that have poorer sexual health than their peers. For example, YMSM account for approximately one third of new HIV infections among men who have sex with men (MSM) in the US [17], with many YMSM reportedly learning about anal sex through direct experience, in the absence of any appropriate sexual education [18]. Importantly, this means that discussion about the higher risks of STIs that are associated with anal sex and potential preventative measures do not occur. YMSM are also more likely to engage in sex while under the influence of drugs (chemsex) meaning there are specific risk factors associated with same-sex behaviour, necessitating targeted and culturally sensitive interventions for this group [19]. Young sexual minority females experience higher rates of forced sex and increased reporting of multiple partners [20], and risk profiles vary extensively based on external variables like the age of their partners [21].

Other examples of high-risk young populations reporting poor sexual health include the finding that young people (aged 15–25) accessing youth mental health services report higher rates of high-risk sexual behaviour and unwanted pregnancy [7] than their peers without mental ill-health. Young people with alcohol or other substance use problems have also been found to be more likely to have multiple partners, less likely to use condoms consistently, and as a result, be at a greater risk for STIs and BBVs [22]. There are similar findings of poor sexual health for youth in other ‘high-risk’ settings such as detention settings and foster care. Youth within detention settings have been found to have a higher incidence of STIs, more sexual partners, and higher rates of pregnancy than youth who have never been incarcerated [23]. Youth in foster care are more likely to be sexually active, more likely to experience sexual debut before the age of 13, and female adolescents to be more likely to experience earlier pregnancy than those who have never been in foster care [24]. Finally, while Evan and colleagues’ review [14] highlighted that black American youth are at an increased risk of experiencing negative sexual health outcomes, ethnic minorities in other countries have also been recognised as having diverse sexual health support needs (e.g., black and minority ethnic youth in London, UK, [25]).

Although there have been significant advances in the treatment of illnesses transmitted through sexual activity, for example the development of Pre-exposure prophylaxis (PrEP) since the HIV epidemic of the 1980/1990s and the Human papillomavirus (HPV) immunisation, there has been markedly less advancement in sexual health promotion interventions. This is particularly so for young people aged under 25, despite the need to promote early intervention during this critical point in life. For example, in the Compendium of evidence-based and best practices for HIV prevention published by the CDC, only 10 of the included 59 interventions were developed for those under 25 [26]. Although there is considerable heterogeneity within this age range (12–25), modern health services are moving towards viewing young people aged between 12 and 25 as a core age range. This makes sense particularly within mental health services, where utilising the typical cut off age of 18 years does not match the developmental trajectory of young people. The same could be said for sexual health. While approaches to treatment are likely to differ within the 12–25-year age range, overall, it represents a time in the life course when sexual health intervention is most critical if we are to see sustainable, long-term positive change in sexual health.

While systematic reviews of interventions for adults within specific high-risk populations have been completed (e.g., mental illness [27,28], homelessness and drug use [29] justice involved individuals [30] and MSM [31]), the same has not for young people other than the review of interventions targeting black American youth [14]. The aim of the current systematic review is therefore to establish what behavioural interventions have been tested to improve sexual health among young people in high-risk populations, and how effective they are.

## 2. Methods

Before beginning the search, review databases (Cochrane, Prospero) were searched to confirm no similar reviews were already completed or in progress, and this review was registered on Prospero and at OSF (ref. 149810; osf.io/ukva9).

### 2.1. Search Strategy

Literature searches were performed on MEDLINE, PsycINFO, Excerpta Medica Database (EMBASE), Cochrane Central Register of Controlled Trials (CENTRAL), Web of Science, and Scopus. Ongoing and unpublished trials were searched for at ClinicalTrials.gov, Australian and New Zealand Clinical Trials Registry (ANZCTR), and the World Health Organization’s ISRCTN registry. The last search was performed on 28 July 2021. Filters were used to limit results to RCTs, and no date limit was set. All search strategies were reviewed by a medical librarian, and are available in Appendix C and the Appendix A.

### 2.2. Inclusion Criteria

The PICO (Patient, Intervention, Comparison, Outcome) framework was utilised to develop our focused inclusion and exculsion criteria (see Table 1).

Additionally, articles were only included if they:Were a randomized controlled trial with ≥2 arms;Reported sufficient data to satisfy PRISMA and Cochrane guidelines for inclusion in the review;Were published in English.

A study was ineligible if:The primary aim of the study was not to address sexual health;There was a later publication of results from the same trial—the paper reporting the longest follow-up data was chosen.

### 2.3. Data Retrieval

Data retrieved were uploaded to the referencing manager Mendeley which was then used to remove duplicates.

### 2.4. Data Screening

Titles and abstracts of papers were screened independently by two reviewers (S.L. and E.B.) following the eligibility criteria, using Covidence online review software. Papers that met eligibility criteria then had full articles screened. Discrepancies between the two reviewers were resolved by a third reviewer (R.G.).

### 2.5. Data Extraction

A standardised form was used to extract data from the eligible trials and included: study citation, place of origin, setting, sample size, intervention type, dose, control, and findings. We noted in the table whether outcomes reported were based on participant knowledge and attitudes (**K + A**), their self-reported behaviour (**B**), was an outcome that was biologically verified, i.e., STI infection or pregnancy (**Bio**), or related to sexual wellbeing, i.e., communication skills or relationship satisfaction (**SW**). A narrative summary of the findings was used to present the data outlined in the data extraction table.

### 2.6. Quality Assessment of Interventions

All articles reviewed were subject to quality assessment using the Cochrane Risk of Bias tool. Assessments were performed independently by two reviewers (S.L. and E.B.) and with discrepancies being resolved by a third party where required (R.G.).

## 3. Results

### 3.1. Search Results

The final searches were performed on 28 July 2021, and 8631 papers were identified in total. The removal of 5968 duplicates left 2663 papers to be reviewed for eligibility, 2461 papers were removed following title and abstract screening, with 174 removed following full text screening. In total, 28 papers from 27 trials met our inclusion criteria with the PRISMA flow diagram (Figure 1) showing details of reasons for exclusions; the most common reason for exclusion was an age range that went beyond 25 years at the beginning of the trial. The 28 included papers were spread over the high-risk groups as follows: four focused predominantly on AOD, four on ethnic minorities, three on homelessness, six on juvenile justice, six on YMSM, two on mental ill-health, three on out-of-home care. Only one trial was identified in LGBTQI+ groups outside of YMSM. The 28 included papers had a combined total of 8312 participants and Table A1 details the intervention delivery methods, outcomes assessed and detailed results of each of the trials, organised by high-risk grouping.

Nearly all (26/27) trials took place in the USA and tested interventions that were delivered in group settings, ranging from 45 min to 8 h, over 1 week to 7 months. Gender differences within each of the trials were minimal and most frequently not discussed within the papers, as reflected in the results reported in Table A1. Four of the interventions were delivered at one time only, and two interventions were delivered entirely via text message. All but two trials took place in locations that the young people were already engaged in (e.g., mental health clinic, group home, drop-in centres), rather than at schools. Some included trials had intersectional populations; where this was the case, they are grouped according to the primary target population as stated by the authors, with secondary groups noted in relevant sections.

### 3.2. Included Trials

#### 3.2.1. Alcohol and Other Drug (AOD) Use

Our review identified four papers describing interventions for youth with AOD-related issues. Two were conducted inside a residential treatment facility [32,33], one in outpatient clinics [34], and one via text message [35]. The two older trials by St Lawrence and colleagues [32,33] were founded on behavioural skills training. The earlier pilot trial [32] only looked at pre-post intervention changes with knowledge, attitudes and behaviour all improving in the intervention group compared to the control. The later full-trial with a 12-month follow-up period reported significant group-time interactions in knowledge, attitudes and behaviour outcomes but poor reporting of, for example, group sizes, means it is difficult to draw strong conclusions. Two more recent trials [34,35] took place in the community which both reported longer-term outcome data (over 12 months), focusing purely on behavioural outcomes. Letourneau and colleagues’ [34] group intervention focused on reducing substance use and sexual risk-behaviours for young people who had been referred to a juvenile drug court. It incorporated caregiver involvement in a contingency management program, including elements of cognitive–behavioural therapy to help teens identify the antecedents of their risk behaviours. In contrast, Suffoletto and colleague’s [35] trial of a sexual risk reduction intervention for female AOD users who attended hospital emergency departments, was delivered via interactive text messages. Neither of these interventions had a significant impact on sexual risk behaviours, with the authors highlighting the challenges of addressing common co-occurring ‘problem behaviours’ with one broad approach and of keeping young people engaged in interventions.

Five other papers identified in this review included young people with substance use problems as secondary groupings, given the commonality of comorbidities between these high-risk groups (Juvenile Justice and Homelessness groups) [36,37,38,39,40]. These will be discussed under their primary grouping. Across all sub-groupings, meta-analyses were not possible due to disparities across outcome measures used.

#### 3.2.2. Ethnic Minorities

We identified four trials that were undertaken in an ethnic minority population outside of black/African-American youths. All four were undertaken in North America with adolescents under 20 years old. One targeted an indigenous population [41] and three had majority Hispanic/Latinx samples [42,43,44]. The trial that targeted a population of American Indian adolescents was provided in the context of a summer basketball camp [41]. This trial of 267 adolescents found significant outcomes regarding knowledge of HIV prevention, and condom use attitudes and self-efficacy, however behavioural outcomes were not measured. Interventions across the three other trials that targeted Hispanic/Latinx samples were also all delivered via culturally sensitive and targeted group settings. Outcomes measured varied considerably making conclusions about what worked best for this target group difficult to draw. Kipke et al.’s trial [43] found improvements in knowledge and attitudes but these did not translate to changes in behavioural outcomes. while Villarruel and colleagues’ trial [44] finding some significant improvement in rates of high-risk sexual behaviours but did not measure knowledge and attitudes. One element of sexual wellbeing (assertiveness and communication skills), was measured by Kipke et al. [43], with the intervention group reporting a significant improve in ability to refuse high-risk, and propose low-risk, behavioural alternatives. One of the most recent trials, Smith and colleagues’ [42] intervention for Hispanic teenage mothers found that both their intervention and control groups improved their condom usage to prevent STIs over time although the intervention did not produce additional change as predicted. Authors suggest a need for involving both genders in condom use decision making with future interventions to include communication skills training around openly discussing condom use. The authors also called for the use of objective outcomes measures of STI infections and pregnancy rates to identify true impact of such an intervention.

Two papers from other high-risk groups also included ethnic minorities, one targeting homelessness [39], and one YMSM [45]. These will be discussed under their primary target group.

#### 3.2.3. Homelessness

Three papers focusing on homeless youth are included in this review; all were conducted at drop-in centres providing other services for homeless youth, and also included content to reduce AOD use [38,39]. The most recent trial by Thompson and colleagues [46] involved piloting a smartphone application in conjunction with two 20 min motivational sessions. They determined that this type of delivery was feasible and had a short-term (2 week) effect on sexual risk behaviours. They concluded a larger trial with a longer follow up was now warranted. Both of the other larger trials only reported significant findings when an unplanned post-hoc analysis was conducted. Slesnick et al. [38] found their intervention only had a significant impact on condom usage when age was factored into the analysis; intervention group participants aged 14–18 used condoms more frequently at follow up than control group youths aged 19–22. While the intervention used by Tucker and colleagues [39] failed to have an impact on knowledge and attitudes, it did decrease unprotected sexual acts for a subsample of their population, specifically participants with multiple sexual partners. Overall, all these trials referred to how difficult it can be to achieve a positive behaviour change in this population but the need to persevere given the multiple risk factors homeless young people often present with and therefore the potential large gains that can be achieved on a public health level. Integrated, engaging interventions that focus on the totality of the young person’s life are likely to be better received given their complex presentations.

#### 3.2.4. Justice-Involved Youth

We identified six papers reporting trials that targeted justice-involved youth [36,37,40,47,48,49]. Four of the six trials took place in detention centres or prisons, while one [47] was conducted in foster care homes that adolescents had been placed in as part of their ‘treatment’. Two of the trials were single gender only (one each for males [48] and females [47]), and two others also included content aimed at reducing drug and alcohol consumption [36,37]. All six trials focused on behavioural outcomes such as condom use, pregnancy and STI incidence. One study [36] reported the reduction in biologically confirmed STI incidence as a result of their sexual risk reduction group with additional content on alcohol and cannabis use. Kerr and colleagues [47], in a trial with girls aged 13–17, found multidimensional treatment foster care reduced the odds of pregnancy over the subsequent 24 months while females in Goldberg et al.’s trial [49] who received a booster session after three months were more likely to use condoms consistently. Two other trials [40,48] did not have a significant long-term impact on risky sexual behaviours although St Lawrence et al. [48] observed significant post-intervention effects for AIDS knowledge, condom use self-efficacy, positive attitudes about condoms and condom use skills.

Four additional papers described three trials where some of the participants were involved with the justice system [34,50,51,52].

#### 3.2.5. LGBTQI+ (Including YMSM)

Within this group, one trial was identified that focused on a LGBTQI+ population, with an additional five trials focused on YMSM [45,53,54,55,56]. As a novel approach to engaging this potentially hard to reach population, two trials were conducted remotely, one of them online [45] and one via text message [55]. The remaining trials occurred in HIV clinics [53,56] and LGBTQI+ community health centres [54,57]. One occurred in Thailand [56] with the remaining trials occurring in North America. All six trials reported behavioural outcomes, typically rates of engagement in protected sex and the number of sexual partners. The recent trial that was undertaken in young people who identified as LGBTQI+ [57] tested a once-off, 3 h, in-person workshop with follow-up text messages for 12 weeks, which aimed to reduce risky sexual behaviours. The author reported higher knowledge and self-efficacy scores and increased uptake of contraception as a result of the intervention.

Of the five trials that targeted YMSM, results overall were mixed with some, but not all, trials reporting a significant improvement in sexual health behaviours for participants who took part in an intervention. The two individual, motivational, interviewing-based interventions both reported significant behavioural findings. Chen et al.’s [53] intervention significantly increased participant’s likelihood of using condoms and Rongkavilit et al.’s [56] intervention reduced frequency of engaging in anal sex but did not significantly improve condom use. Two of the five trials [45,54] also reported knowledge and attitude outcomes, however neither of these interventions produced significant changes in these outcomes. The two trials not undertaken face-to-face, [45,55] appeared to have limited impact on knowledge, attitudes and behaviour in their current form. Mustanski and colleagues [45] concluded that an appropriately powered trial of their online intervention was required to truly establish intervention effects.

#### 3.2.6. Mental Ill-Health

Two papers targeting young people with mental ill-health were identified; one recruited from mental health outpatient clinics [58], and the other [59] recruited high school students with “emotional or behavioural problems”. Both were three-arm trials and produced mixed results. In Brown and colleague’s 2017 study [59], the HIV prevention plus affect management (AM) intervention appeared to have more of an effect on sexual behaviours than a skills-based HIV prevention (SB), although both active interventions significantly improved HIV knowledge and condom attitudes at six months follow-up. There was no impact on engaging in sexual intercourse with concurrent substance use in either arm of the trial. The other trial [58] tested two HIV prevention interventions, one being family-based, and the other adolescent-only. Both significantly improved sexual behaviours at three months compared to control, but the family-based intervention also improved HIV knowledge and parent–teen sexual communication.

Two papers included in other high-risk groups also address youth with a history of abuse and/or mental ill-health [51,60], again highlighting the common comorbidities that these at-risk young people frequently experience.

#### 3.2.7. Out-of-Home Care

Three papers were identified from two trials that focused on delivering an intervention to young people in out-of-home care [50,51,52], with both trials recruiting participants that were also involved in the juvenile justice system. The two trials tested intensive (at least twice weekly) group-based interventions delivered in groups within the out-of-home care setting. Whilst the older trial [51] focused on HIV/AIDS prevention, Green and colleagues’ [50] trial focused on pregnancy as well as HIV and STI prevention. Both trials found long-term (up to 12 months post intervention) significant change in knowledge and attitudes, but non-significant changes in sexual behaviours. Green et al. [50] also assessed the sexual wellbeing concept of ‘ability to communicate with partner’ with the intervention group reporting a significantly higher ability over the 12 months follow up period.

### 3.3. Quality Assessment

The Cochrane Risk of Bias tool included in Covidence online review software was used to assess the quality of all included papers, see Figure 2 and Figure 3. Figure 2 shows that there was a high or unclear risk of bias for nearly all trials in the blinding of participants or personnel domain. The only domains where over half of the included trials were at a low risk of bias were incomplete outcome data and selective reporting. The use of ‘computer assisted self-interviewing’ techniques for data collection meant that blinding of outcome assessments was possible in a third of trials. Poor reporting across many papers meant that risk of bias was unclear for a large proportion of the trials.

## 4. Discussion

This systematic review of behavioural and psychosocial interventions to improve the sexual health of young people in high-risk groups identified a total of 28 papers from 27 trials published between 1993 and 2021. The high-risk groups with the most trials reported were in justice-involved youth, AOD and YMSM. Unfortunately, given the lack of commonality of outcome measures utilised within each of the high-risk populations, it was not possible to undertake meta-analyses. Only three trials were identified in OOHC and homeless populations, and only two trials in mental ill-health. These findings highlight the areas where considerable work is still required. Whilst five trials of YMSM were identified, only one focused on broader LGBTQI+ groups, highlighting a considerable lack of depth and replication in this group. The numbers of trials completed in these high-risk groups of young people was considerably lower than in adult populations [27,28,29,30,31]), with the widest gaps being in mental ill-health (2 in youth, 13 in adults [28]) and in individuals in the justice system (5 in youth, 27 in adult, [30]). While interventions varied substantially in regards to delivery method, the most common element across all of the trials identified was that they were conducted in locations that the high-risk young people were already accessing support which is in contrast to findings by Evans and colleagues [14] supporting the delivery of interventions to young Black Americans in school settings.

The most frequent outcomes reported in the included papers related to self-reported measures of behaviours that lowered risk of HIV and other STIs, such as frequency of unprotected sex, consistent condom use, and number of sexual partners. Change in knowledge and attitudes as a result of the intervention was also frequently measured. Most of the interventions were based on a model of health behaviour change, such as the Information–Motivation–Behaviour skills model [61,62], which implies that whilst information is important for health behaviour change (i.e., knowledge), an individual’s own motivation to change is also critical (i.e., attitudes). This is a typical model on which sexual health promotion interventions are based and shown to be effective in adult populations [27,30]. This review identified a number of trials [43,48,50,51] that demonstrated that knowledge and attitudes toward sexual health and risk taking could be significantly improved in the intervention arm, but this did not translate into self-reported changes in behaviour. This was frequently attributed to the complex presentations of the participants that the interventions targeted, making the translation of knowledge into behaviour change an even greater challenge within such public health interventions. Sexual health behaviour change requires assertiveness, planning, access to resources such as condoms and contraception and engaging with partners that are non-coercive [13,63]. For some groups this can be especially challenging when they face isolation, stigma, exploitation and abuse [63,64]. Modern sexual health promotion interventions should consider the need for empowerment, communication and social skills training and access to appropriate resources and support.

A minority (3/26) of the trials assessed outcomes relating to ‘sexual wellbeing’. In those three trials, the only parameter of sexual wellbeing included was communication skills with partners or parents. Other critical elements of taking a ‘sexual wellbeing’ approach such as sexual pleasure and sexual justice [65,66] were not seen in the trials we identified. There remains a need for all sexual health promotion interventions to encompass healthy relationships as well as promoting the ability to communicate needs and boundaries, and negotiate contraceptive use, without solely focussing on the prevention of negative outcomes [65,67]. There is increasing recognition that taking a positive ‘sexual wellbeing’ approach can reduce risk by increasing ownership and agency [68,69]. Given that young people in high-risk groups have often experienced considerable societal and psychosocial challenges that have diminished their sense of agency and control in life [70,71,72], applying this positive approach to interventions for these individuals is even more critical.

Modern trials in this area could also be strengthened by including objective outcome measures of sexual health through the use of biological outcome measures such as pregnancy and STI tests rather than relying on subjective self-report measures. Out of the 26 trials we identified, only one carried out STI testing as an outcome measure. By doing so, policy makers, service providers and clinicians could ensure their efforts to improve the sexual health of young people in high-risk groups are grounded in the best available evidence. This is a similar conclusion drawn by the systematic review completed by Evans and colleagues [14] in which they propose further meta-analyses should be completed when further studies have evaluate the effectiveness of interventions on biological outcomes. With this shift in outcome measure use, the long-term success of preventative sexual health programs can be truly evaluated.

### 4.1. Implications and Applicability

When considering how these results could be used to inform changes to policy development, workforce training, health promotion strategies and treatment care plans, it is important to emphasise the need for holistic, accessible and compassionate treatment. While improvements to knowledge and attitudes toward sexual health were present in many of the interventions included in this review, we found that they often did not necessitate any significant changes in sexual health behaviours. A one-off group intervention, for example, may improve condom use in the short term but further reminders, check-ins and discussion is likely needed if these changes are to be sustained in the long-term. Nearly all trials took place in locations that the individuals were accessing outside of school environments. This could speak to supporting an individual’s sexual health needs ‘in-house’ at the community services they are already accessing, rather than referring out to different primary care settings [73,74].

The frequent intersectionality of the populations identified in this systematic review could mean that young people struggle to access the support they need. When an individual is a member of several vulnerable groups, they are at risk of slipping through the cracks due to a lack of ‘ownership’ for the issue of their sexual health. Breaking down the siloed nature of physical health, mental health and social care services for these vulnerable individuals remains a key task at a clinical and policy level. In particular, finding novel, effective ways to engage these groups, and supporting them to consider and address their sexual health at the critical period of adolescence. Early, effective intervention for vulnerable youth could prevent the exacerbation of problematic economic and health disparities. For example, in the UK, considerable success has been seen in public health interventions that target teen pregnancy, however there are still large disparities between certain groups [75].

As is evident from this systematic review, there are a lack of trials completed in relation to intersectional populations. Additionally, although ethnic minorities have been represented in some manner here, there are many more aspects of catering sexual healthcare to CALD young people which needs further attention (e.g., young women who have experienced FGM). The recognition of high-risk groups without further action to minimise said risk could contribute to poor sexual outcomes for society as a whole, and this review highlights the high-risk groups that could be prioritised.

In addition to helping to break down stigmas and decrease barriers to accessing care, work needs to be done in order to assist young people to empower themselves to achieve a healthy and fulfilling sexual life. This should encompass both good sexual health (e.g., being offered vaccinations and regular screening) and sexual wellbeing (socially, emotionally and psychologically positive sexual experiences). A starting point to begin changing the societal norms surrounding sexual health, could be to integrate more sex-positive and practical sex education in schools [68,69].

### 4.2. Limitations

The results of this systematic review need to be considered in light of several limitations both of our review processes and of the evidence included in the review. Firstly, we excluded trials that utilised an age range of up to 29 years old, meaning we may have missed key interventions due to differences in definitions of ‘young people’. Second, the outcome measures used across studies varied considerably meaning it was not possible to complete meta-analyses of the data. Even when outcome measures were the same, the specific questions asked often differed, for example the time frame participants were asked to report on. Future research in the area could be strengthened by an international consensus on how to collect data on high-risk sexual behaviours [76]. Third, a lack of consistency across delivery method meant strong conclusions were difficult to reach given the small study numbers. Finally, the applicability of our findings for all groups, but particularly for ethnic minorities populations is limited by finding only trials from North America. This suggests that we cannot reliably extrapolate to other minority groupings such as Culturally and Linguistically Diverse (CALD) populations in Australia, or Black, Asian and Minority Ethnicities (BAME) in the UK and further tailoring of content to specific cultural groups is likely required. This cultural adaptation should preferably be co-produced with those who represent the voice of those cultures, before they are tested in other settings and groups.

## 5. Conclusions

This systematic review found some support for sexual health promotion interventions that were tailored to specific high-risk populations of young people. Being unable to perform meta-analyses of these trials identified suggests that the field remains underdeveloped, and more randomised trials are required. Tackling the comorbidities commonly encountered by these populations is a challenge that requires consideration when developing or modifying interventions for these vulnerable populations. Attention should be paid not just to the health behaviour outcomes such as condomless sex, but also to the quality of the sexual relationships that young vulnerable people engage in, as well as the contexts in which sex occurs. This includes non-consensual sex and assault, exploitation, coercive controlling behaviours, lack of self-esteem and lack of social skills such as assertiveness, condom refusal in partners and intimate partner violence.

Vulnerable young people experience significant challenges in their sexual health and wellbeing. As we have seen through teenage pregnancy interventions [75], it is likely that a good understanding of the specific issues faced must be factored into tailored and co-produced interventions that recognise the holistic nature of sexual wellbeing. Sexual wellbeing is not merely the prevention of disease, but also the empowerment of disenfranchised young people by focusing on building knowledge, a sense of self-worth and promoting positive, rewarding sexual relationships that are free from abuse or coercion.

## Figures and Tables

**Figure 1 ijerph-18-09063-f001:**
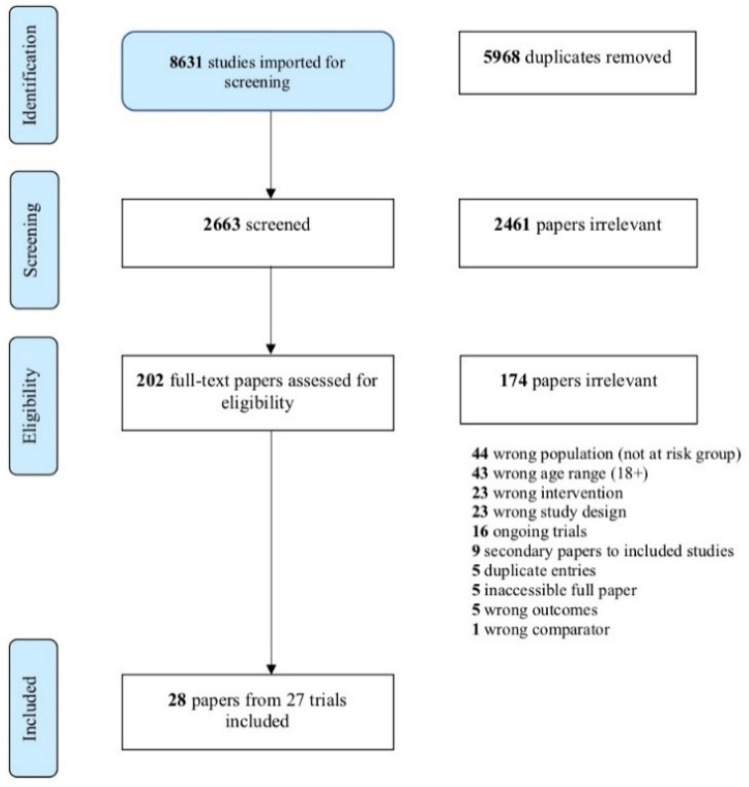
PRISMA flow diagram of included studies.

**Figure 2 ijerph-18-09063-f002:**
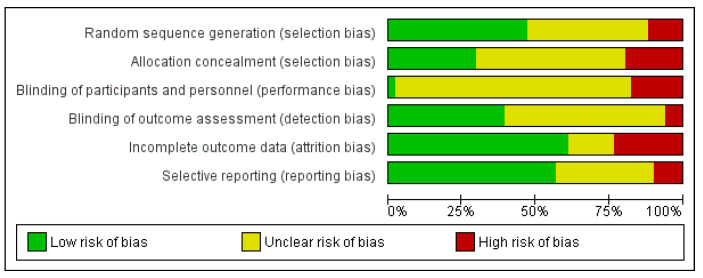
Risk of bias graph: review authors’ judgements about each risk of bias item presented as percentages across all included studies.

**Figure 3 ijerph-18-09063-f003:**
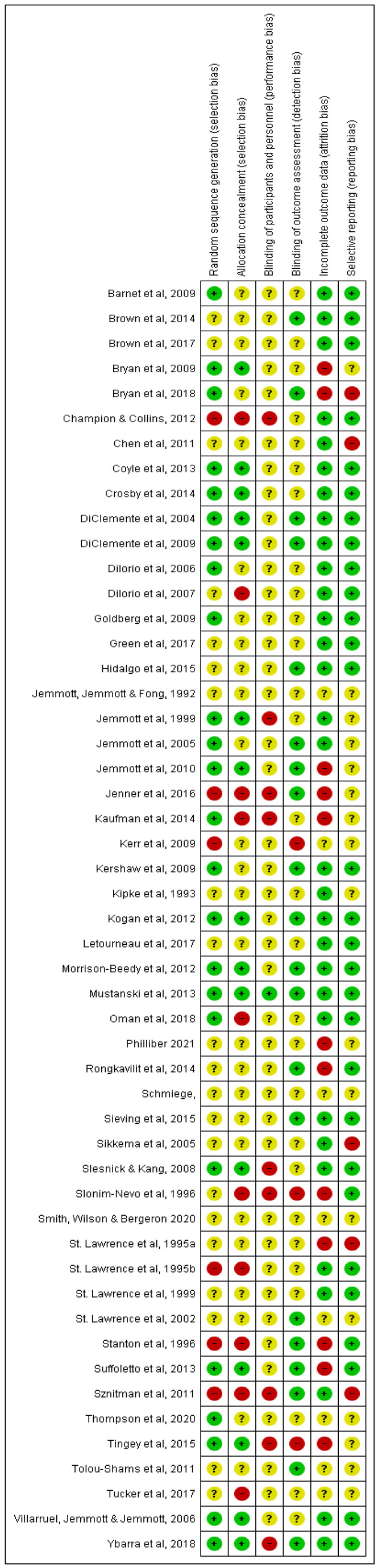
Risk of bias summary: review authors’ judgements about each risk of bias item for each included study.

**Table 1 ijerph-18-09063-t001:** PICO inclusion criteria.

PICO Element	Inclusion Criteria	Exclusion Criteria
Population	Young people ≤25 years old, of any gender, who are members of one of the following high-risk groups: Alcohol and other drug use (AOD) Ethnic minority (min. 75% of participants) Homeless Justice-involved youth LGBTQI Mental ill-health Out-of-home care (OOHC)	Samples including participants >25 years old at start of trial If the trial targeted black American youth as the only ‘high-risk’ criteria given previously completed review in this population.
Intervention	Any psychosocial or behavioural intervention aimed at promoting sexual health and/or sexual safety-taking behaviours	Any intervention that targets parents of young people rather than the young people directly, or any intervention that is focused solely on promoting abstinence
Comparison	Any non-pharmacological comparator (e.g., waitlist control)	A pharmacological comparator
Outcome	Sexual health or safety-taking behaviours, e.g., condom use; Biological indicators of sexual safety-taking behaviour, e.g., STI incidence, unwanted pregnancy; Changes in knowledge of, or attitudes toward, sexual health and safety. Measures of sexual wellbeing e.g., communication skills or relationship satisfaction	Abstinence

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
