# Peer review of "Improving the Sexual Health of Young People (under 25) in High-Risk Populations: A Systematic Review of Behavioural and Psychosocial Interventions"

_ijerph, 2021, doi:10.3390/ijerph18179063_

Round 1
Reviewer 1 Report
Brown et al in this paper, “Improving the sexual health of young people (under 25) in high-risk populations: a systemic review of behavioral and psychosocial interventions, sought to perform system analysis of twenty-six peer reviewed reports from 25 randomized controlled-trials. Authors used elaborate and descriptive “inclusion criteria” for the studies to be analyzed in this system review. They summarize well and highlight the need for targeted interventions and research to improve the sexual health of the minority populations at risk. However, reviewer finds the paper unacceptable for publication in its current form. The key reasons are listed below:
- The intervention methods and outcome measured across studies varied considerably making it difficult to come to common conclusions which authors acknowledge in “limitation” sections. It is imperative that the review provides basic outline of what intervention strategies works for which population and parameters. The review must show path forward.
- Importantly, many studies listed in result section lacked clear “intervention studies”. Were they mentioned in original paper? Many do not include behavioral interventions which was the objective of this review. How do readers know what interventions were used in those trials and what parameters of sexual health were improved?
- Authors included ethnic minorities together with population with homeless-ness, alcohol and drug abuse, justice-involved youth, mental and behavioral health concerns and LGBTQI+. Was it only because those were ALL the studies that satisfied inclusion criteria? It is reviewer’s understanding that the underlying difference in behavioral and sociopsychological status of these subpopulations prevents studies from analyzed together. Reviewer likes to think that authors agree as they suggest a targeted trauma-based approach for LGBTQIA+ population (Line 339).
- Authors discuss significant impact on outcome measures in several of the studies. However, statistical measures are not mentioned once. If available, authors are encouraged to discuss those. In a study that involved homeless population, the participants are 14-18 years old, control were 19-22 YO. Is control population also a homeless? If so, why is that? Will that have any effect on outcome measures? The 14- to 18-year-old are barely a consenting age and relatively less mature, experienced, and self-efficacious which may have impact on behavioral outcomes. Will this impact how the outcomes measures were different (if), in some studies that involved family-based or caretaker-based interventions?
- Many studies do not mention the age group or gender of the population outright. Was that information available? Do gender play role?
- In most studies, the outcomes measure for sexual wellbeing is discussed only in-terms of participants ability to freely communicate to their partner. Was that the only parameter used in those studies. It may have to do with the fact that many studies included in the analysis are not recent. Older studies, researcher and policymakers failed to view sexual well-being in context of sexual health, sexual pleasure, and sexual justice. The four pillars for a comprehensive public health approach to sexuality should be viewed with integrated approach.
- Overall, reviewer finds that there is huge disconnect between objective of this systemic review, the studies included, and the conclusions reached. For instance, authors suggest a “one-shop-stop” as an effective alternation to primary care setting but fail to acknowledge that most studies in this review are population mostly probably not in “school environment”.
Author Response
Thank you for your timely review of our paper. We found the reviewer comments very helpful in improving our paper further and hope that we have adequately addressed the comments in our responses below
The intervention methods and outcome measured across studies varied considerably making it difficult to come to common conclusions which authors acknowledge in “limitation” sections. It is imperative that the review provides basic outline of what intervention strategies works for which population and parameters. The review must show path forward.
Response: Thank you for this reflection, we agree and have made substantive edits to the results section to strengthen our narrative synthesis. Please see track changes in the document that has been resubmitted for these.
Importantly, many studies listed in result section lacked clear “intervention studies”. Were they mentioned in original paper? Many do not include behavioral interventions which was the objective of this review. How do readers know what interventions were used in those trials and what parameters of sexual health were improved?
Response: Apologies for the lack of clarity, the details outlined in this comment can be found in Table A1 and confirms that the interventions collated in this review are all behavioural and/or psychosocial focused. In addition, we have made edits throughout our results to clarify the interventions used and sexual health parameters in each grouping.
Authors included ethnic minorities together with population with homeless-ness, alcohol and drug abuse, justice-involved youth, mental and behavioral health concerns and LGBTQI+. Was it only because those were ALL the studies that satisfied inclusion criteria? It is reviewer’s understanding that the underlying difference in behavioral and sociopsychological status of these subpopulations prevents studies from analyzed together. Reviewer likes to think that authors agree as they suggest a targeted trauma-based approach for LGBTQIA+ population (Line 339).
Response: As detailed on page 2 of the manuscript, we derived our definition of ‘youth at risk’ from the Australian Department of Health topic resources. We felt that by reviewing these at-risk populations together we would better understand what research had been undertaken in each of these areas, including the commonalities and differences between these groups. While the groups share some psychosocial and cultural challenges, these are of course unique to each population, and if we attempted to analyse the finding together in a meta-analysis, we would face criticism that the groups are too different to compare results across therefore must be analysed separately. We hope that in time, we will be able to perform meta-analyses on these data as the publication of new trials come out and the outcome measures that are used are streamlined. We make further reference to this in our limitations section (line 495) as well as edits to our abstract.
Authors discuss significant impact on outcome measures in several of the studies. However, statistical measures are not mentioned once. If available, authors are encouraged to discuss those. In a study that involved homeless population, the participants are 14-18 years old, control were 19-22 YO. Is control population also a homeless? If so, why is that? Will that have any effect on outcome measures? The 14- to 18-year-old are barely a consenting age and relatively less mature, experienced, and self-efficacious which may have impact on behavioral outcomes. Will this impact how the outcomes measures were different (if), in some studies that involved family-based or caretaker-based interventions?
Response: We have chosen to contain the statistical findings to our table A1 and keep the synthesis of the trials in the results section as narrative to keep things clearer for the reader. We are happy to edit these sections to include the precise statistical measures if required by the editors.
Regarding the paper on homelessness, we believe the reviewer is referring to Slesnick and Kang (2008), but they have not interpreted the age groups correctly. Intv group has a mean age of 19.03 while control group is 19.40 years. The authors of this paper performed post hoc analysis to compare the two age groups in each trial arm. We have clarified in table A1 that this statistical analysis was performed post hoc by adding the following to the final column: “Post hoc analysis revealed”.
Many studies do not mention the age group or gender of the population outright. Was that information available? Do gender play role?
Response: In table A1 we detail the age group of the population in the study, and where samples are only one gender, we identify this in the study population column. It is worth clarifying for readers that gender difference within the samples are minimal and are not focused upon in the studies included in the review and we have clarified this in section 3.1, line 205.
In most studies, the outcomes measure for sexual wellbeing is discussed only in-terms of participants ability to freely communicate to their partner. Was that the only parameter used in those studies. It may have to do with the fact that many studies included in the analysis are not recent. Older studies, researcher and policymakers failed to view sexual well-being in context of sexual health, sexual pleasure, and sexual justice. The four pillars for a comprehensive public health approach to sexuality should be viewed with integrated approach.
Response: Thank you for this insightful reflection, we agree and have amended and added to the following to the discussion to refer to this (beginning line 358):
“A minority (3/26) of the trials assessed outcomes relating to ‘sexual wellbeing’. In those three trials, the only parameter of sexual wellbeing included was such as communication skills with partners or parents. Other critical elements of taking a ‘sexual wellbeing’ approach such as sexual pleasure and sexual justice [61, 62] were not seen in the trials we identified. Including this focus recognises There remains a the need for all sexual health promotion interventions for young people in high-risk groups to encompass healthy relationships as well as promoting the ability to communicate needs and boundaries, and negotiate contraceptive use, without solely focussing on the prevention of negative outcomes [61, 63]. There is increasing recognition that taking a positive ‘sexual wellbeing’ approach can reduce risk by increasing ownership and agency [64, 65]. Given that young people in high-risk groups have often experienced considerable societal and psychosocial challenges that have diminished their sense of agency and control in life [66-68], applying this positive approach to interventions for these individuals is even more critical.”
Overall, reviewer finds that there is huge disconnect between objective of this systemic review, the studies included, and the conclusions reached. For instance, authors suggest a “one-shop-stop” as an effective alternation to primary care setting but fail to acknowledge that most studies in this review are population mostly probably not in “school environment”.
Response: Apologies for the confusion in our text, we agree that the point that our populations are not accessing school and that is why they should access care at the services they are already attending. We have edited the following statement beginning on line 389 for clarity:
“Nearly all trials took place in locations that the individuals were accessing outside of school environments. This could speak to supporting an individual’s sexual health needs ‘in-house’ at the community services they are already accessing, rather than referring out to different primary care settings.”
We have removed reference to the ‘one stop shop’ in line with comments from reviewer 2.
Reviewer 2 Report
Dear authors,
Your manuscript is interesting but I need you to answer some questions:
METHODS
Search Strategy:
- Authors must specify what the search equation was like and how they combined the Boolean operators. The research must be reproducible.
- WOS is not a database, but a meta-search engine.
Inclusion Criteria:
- In the inclusion/exclusion criteria there is information that the authors have not specified. The study is not replicable.
RESULTS
- Authors should do a reverse search to get more results.
DISCUSSION
- You cannot use the references of the "results" in "introduction" or "discussion". This is wrong.
- There are few references in "discussion".
- The authors do not explain their results well with references to other studies.
Implications and Applicability:
- This section is very long and generally does not use references.
REFERENCES
Many bibliographies are obsolete and some citations are incomplete. The bibliographic citations used are more than 5 years old (69,4% not including "results"). Authors should update the "introduction" and "discussion" references.
Author Response
Reviewer 2
Thank you for your timely review of our paper. We found the reviewer comments very helpful in improving our paper further and hope that we have adequately addressed the comments in our responses below.
METHODS
Search Strategy:
- Authors must specify what the search equation was like and how they combined the Boolean operators. The research must be reproducible.
Response: We have detailed one of our search strategies in appendix B of the document and created a supplementary material document that details the other search strategies.
- WOS is not a database, but a meta-search engine.
Response: We agree this is the case, as is the same for Scopus and believe that the text we have in the document on page 3 (beginning line 144) clarifies this:
“Literature searches were performed on MEDLINE, PsycINFO, Excerpta Medica Database (EMBASE), Cochrane Central Register of Controlled Trials (CENTRAL), Web of Science, and Scopus”. We are happy to edit wording further if you feel this is necessary.
Inclusion Criteria:
- In the inclusion/exclusion criteria there is information that the authors have not specified. The study is not replicable.
Response: We have re-reviewed our text and are unsure what information the reviewer feels is missing from the inclusion/exclusion criteria. If this can be clarified we will happily make the edits.
RESULTS
- Authors should do a reverse search to get more results.
Response: We performed multiple comprehensive searches for this review, including searching for articles recently published from eligible trials identified in trial registries. We are therefore confident that our search strategies identified all eligible papers out there.
DISCUSSION
- You cannot use the references of the "results" in "introduction" or "discussion". This is wrong.
Response: We have reviewed the introduction section and double checked that we do not refer to our results in this section. Referring to our results within our discussion is of course appropriate and we are not sure what issue the reviewer takes with this, if they wish to clarify we will happily make the required edits.
- There are few references in "discussion".
Response: Apologies for this omission, we have added in references throughout our discussion to support our arguments.
- The authors do not explain their results well with references to other studies.
Response: We have added in references throughout our discussion to support our arguments.
Implications and Applicability:
- This section is very long and generally does not use references.
Response: As clinician-researchers ourselves, we feel that the implications and applicability section to be one of the most useful parts of a research paper, however we have made a number of edits to this section to shorten it as well as adding in references throughout.
REFERENCES
Many bibliographies are obsolete and some citations are incomplete. The bibliographic citations used are more than 5 years old (69,4% not including "results"). Authors should update the "introduction" and "discussion" references
Response: Thanks for picking this up but upon review of our references we are not sure which citations are not complete. We have used endnote reference management software and the reference style of the journal to ensure that references match the journal requirements. In regards to citations being over 5 years old, we are not sure which articles the author is referring to in particular but we feel that it is important to prioritise quality of an article referenced over age of article. If the reviewer wishes to clarify which references they feel need updating we would be more than happy to make these edits.

Reviewer 3 Report
This is a systematic review article examining the RCT interventions to improve sexual health outcomes among high-risk young people under the age of 25. The paper was well written. The authors did a good job explaining their objectives, rationale, and the process of their systematic review. The results section was clearly presented by the specific high risk group.
However, one major issue was that the results were presented in a more additive nature, i.e., A study found this, while B study found that. While a summary of previous interventions were important, what the readers (or at least myself) look for in a systematic review was probably more about what worked, what didn't work, and why. I think for each group, it would be critical that that the authors add some content on a coherence narrative for the interventions of that group to describe previous work, strengths and limitations, and future directions.
Round 2
Reviewer 2 Report
Dear authors,
Thanks for your reply. I remind you that 60% of "discussion" references are older than 5 years and this section is short. In the "discussion" it is necessary to compare the "results" with other current studies. Therefore current references should be used.
Otherwise, the explanations of the authors are satisfactory. The paper has greatly improved its quality.
Congratulations on your work.
Best regards
Author Response
Thank you for re-reviewing our paper, we have made the following additional edits to the discussion section in response.
We have re-reviewed PRISMA guidelines for the discussion section, including the recent publication of “PRISMA 2020 explanation and elaboration: updated guidance and exemplars for reporting systematic reviews”. This includes the following headings:
“Item 23a. Provide a general interpretation of the results in the context of other evidence.”
To ensure that we have addressed this we have added several sections to our discussion that links the findings in the current review back to the findings from previous reviews in adult populations, referenced in the introduction (refs 27-31) and the one systematic review in the area in youth (Evans et al 2020).
Items 23b and c refer to any limitations of the review processes used and of the evidence included in the review which we have added a couple of comments to, to ensure we adequately address both of those limitation requirements.
Finally, “item 23d. Discuss implications of the results for practice, policy, and future research” is outlined in section 4.1
Regarding the age of references in discussion section, we have added some newer references and also re-reviewed other recent systematic reviews, for example Evans et al 2020, and believe that the profile of the age of our references in our discussion section is similar to theirs; that is, nearly all are less than 10 years old (when you exclude the trials from our review that we reference). Older papers are seminal works that we feel require the original source to be referenced.
We hope these edits address your remaining concerns and we thank you once again for your time reviewing our paper.